# Geochemical Characteristics of Tailings from Typical Metal Mining Areas in Tibet Autonomous Region

**Rengui Weng** [1], **Guohong Chen** [1], **Xin Huang** [1], **Feng Tian** [1], **Liufang Ni** [1], **Lei Peng** [1], **Dongqi Liao** [1,*] **and Beidou Xi** [2,*]

[1] College of Ecological Environment and Urban Construction, Fujian University of Technology, Fuzhou 350118, China; wengrengui109@fjut.edu.cn (R.W.); 13052696442@163.com (G.C.); huangx_fal@163.com (X.H.); windt8023@163.com (F.T.); niliufang@fjut.edu.cn (L.N.); fuzhoupl2008@163.com (L.P.)

[2] Fujian Eco-Materials Engineering Research Center, Fujian University of Technology, Fuzhou 350118, China

[*] Correspondence: ldq0792@163.com (D.L.); 19821499@fjut.edu.cn (B.X.)

**Abstract:** With the exploring and developing of mineral resources in the Tibet Autonomous Region for many years, a large number of tailings have been produced. A total of 17 tailings samples from borehole cores were collected from different tailings ponds in different regions of Tibet. The results showed that the mineral composition and content of tailings in each research area were different. Among them, quartz was the most abundant mineral in most tailings. The major elements of tailings mainly included Si, Al, Fe, Mg, Mn, Ca, Na, K and so on. S existed in different types of tailings. In the analysis of trace element composition, it was found that the content of some elements had approached the lowest industrial grade, which has potential recycling value, such as Mn, Zn, Pb and P. Through the detection of radioactive elements (Ra-226, Th-232 and K-40), it was shown that there were great differences among different types of tailings, and their different contents would bring potential hazards to the safety of the surrounding environment and human health. Similar results were found in the analysis of particle size characteristics of tailings. These results are of great significance for the future utilization and resource utilization of tailings pond.

**Keywords:** Tibet Autonomous Region; metal tailings; mineralogy; element composition; particle size characteristics

## 1. Introduction

Tailings have become the industrial solid waste with the largest output and stockpile in China. There are more than 12,000 tailings ponds, and the cumulative stock of tailings has exceeded 15 billion tons in China [1]. A large number of tailings would not only occupy the land, but also easily lead to environmental pollution and geological disasters [2]. Due to the differences in beneficiation processes in different mining areas, tailings contain many valuable resources, especially some rare dispersed elements, which have high economic value [3]. By utilizing these resources well, the resource shortage problem can be alleviated, simultaneously having a positive impact on the protection of the land and environment.

Tibet Autonomous Region is located in the southwest frontier of China. It is located in the main body of the Qinghai Tibet Plateau, covering an area of more than 1.2 million km$^2$. The average altitude of the whole region is more than 4000 m. It belongs to the low temperature and hypoxia area of the plateau. According to decades of geological exploration, Sanjiang metallogenic belt, Yajiang metallogenic belt, Ban gong Nu jiang metallogenic belt and Gangdisi metallogenic belt have been found in Tibet. Mineral resources are relatively complete and widely distributed [4]. According to statistics, 101 kinds of minerals have been discovered in Tibet, with more than 2000 mineral sites. Copper and boron mines have a resource base for building large and medium-sized mines; iron ore, lead-zinc ore and magnesite have shown resource prospects that can be developed on a large scale [5]. The

country's scarce chromium and copper minerals have not only large reserves in Tibet, but also high grades. The grade of chromite is as high as about 50%. The proven prospective reserves rank first in China. In the "Qinghai Tibet project" completed by China Geological Survey Bureau, 32 large-scale and super large deposits were newly discovered, and 7 super large and 25 large deposits such as Qu long, Jia ma and Xiong cun were found [6]. New resource reserves included 31.94 million tons of copper, 15.19 million tons of lead and zinc, 569 tons of gold and 23,015 tons of silver, with a potential economic value of CHN 2.7 trillion [7]. Tibet would be expected to become China's largest resource reserve base. In addition, Tibet is rich in copper resources. Among the top ten copper mines in China, six are in Tibet, and the top four are in Tibet. At the same time, Tibet has great potential for copper resources, which was expected to reach more than one-third of the country. It was mainly distributed in Gang di si metallogenic belt, Yulong metallogenic belt in eastern Tibet and Ban gong hu Nu jiang metallogenic belt [4]. Due to the diverse types of non-ferrous metals in China, the complex metallogenic geological background and genetic types of ore deposits, as well as the differences in mineral processing technology and level, the element content of tailings produced by various non-ferrous metal mines vary greatly. Due to the diverse types of non-ferrous metals in China, the complex metallogenic geological background and genetic types of ore deposited, as well as the differences in mineral processing technology and level, the element content of tailings produced by various non-ferrous metal mines varies greatly.

Regarding most of the research and investigation of the tailings, the focuses are mainly on the recovery and utilization of tailings, the impact on the surrounding environment and ecological restoration, the activation and migration of heavy metallic elements, the formations of acid water (AMD) and secondary minerals [8–10]. However, there are few reports on the overall and systematic geochemical investigation of elements in different metal tailings ponds, especially in Tibet. It is not conducive to the potential resource evaluation and subsequent comprehensive utilization of the tailings pond. This study is the first to explore the elemental geochemical characteristics of tailings samples from different tailings ponds of typical metal mining areas in the Tibet Autonomous Region. The mineral composition, chemical composition, elemental composition and particle size of tailings samples from five different types of tailings ponds were analyzed.

## 2. Site Description

A total of 17 tailings samples from borehole cores were collected from different tailings ponds in different regions of Tibet. The typical metal mine tailings included: chicken male village molybdenum tailings song water county in Tibet, Lianda Jin mining co., LTD gold concentrator tailings, Yexin mining co., LTD., rapp iron ore tailings, evening rhett township mining company card transcribing contour lead-zinc mine tailings and Tibet dragon copper industry co., LTD., maizhokunggar county dragon copper polymetallic concentrator tailings displacement of five typical metal mine tailings. Location of tailings from typical metal mining areas and sampling sites were shown in Figure 1.

The five typical tailings ponds selected were molybdenum ore, gold ore, iron ore, lead-zinc ore and copper ore tailings. The specific location, altitude, longitude, dimension, temperature and humidity of the tailings at specific samplings are shown in Table 1. It can be seen from Table 1 that all the samples taken belonged to low temperature and hypoxia areas of the plateau. This paper mainly introduces the specific location distribution, actual sampling sites and special ecological environment of typical metal mining areas in the Tibet Autonomous Region and emphasizes that the Tibet Autonomous Region belongs to a low temperature and hypoxia plateau region.

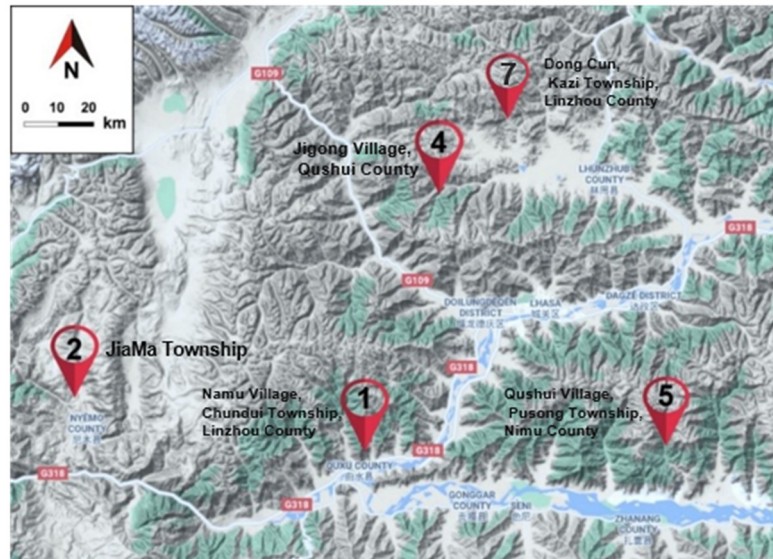

**Figure 1.** Location of tailings from typical metal mining areas and sampling sites in Tibet Autonomous Region (Sampling point 1 is iron ore tailings; Sampling point 2 is copper mine tailings; Sampling point 4 is molybdenum mine tailings; Sampling point 5 is gold mine tailings; Sampling point 7 is lead-zinc mine tailings).

**Table 1.** Sampling situation of typical metal tailing pond in Tibet Autonomous Region.

| Type | Address | Temperature | Humidity | Altitude | Longitude | Dimension |
|---|---|---|---|---|---|---|
| Iron ore tailings (1#) | Namu Village, Chundui Township, Linzhou County | 10 °C | 91% | 4400 | 91.05 | 30.01 |
| Copper mine tailings (2#) | JiaMa township | 12 °C | 79% | 4010 | 91.39 | 29.42 |
| Molybdenum tailings (4#) | Jigong Village, Qushui County | 17 °C | 55% | 3668 | 90.77 | 29.39 |
| Gold mine tailings (5#) | Qushui Village, Pusong Township, Nimu County | 23 °C | 31% | 4081 | 90.13 | 29.50 |
| Lead and zinc tailings (7#) | Dong Cun, Kazi Township, Linzhou County | 10 °C | 90% | 4330 | 90.91 | 29.86 |

Note: number # represents the sample number.

## 3. Materials and Methods

### 3.1. Tailings Sampling

In total, 17 tailings samples were collected from 17 drill cores in September 2020 from the typical metal mining areas in the Tibet Autonomous Region. Drilling machines (diesel rigs) were used to collect tailings cores. In all cases, the drill cores reached the underlying subsoil. Samples were collected at 1 m intervals, the sampling depth was 10 m and each sample weighed between 2.5 kg and 3.0 kg. The density and moisture content of the tailings were measured by using a ring sampler on site. The samples were dried at 105 °C, then pulverized to grain size <0.074 mm (this grain size was considered to ensure the homogeneity of the sample), homogenized and divided into bottles for analysis.

### 3.2. Mineralogy

Five samples were selected from five different tailings cores for particle-size testing with a BT-9300s laser particle-size analyzer. Different samples were studied with an optical microscope (OLYMPUS BX51) under transmission and reflected light and a Rigaku D/max 2550 X-ray powder diffractometer at Shanghai University with Cu Kα radiation at 40 kV and 40 mA. Polished and thin sections of the tailings samples were prepared by standard techniques [11].

### 3.3. Geochemistry

Five different samples were selected from different cores for pH determination according to the SFS-ISO 10390 standard test method with a Thermo Orion 4-Star pocket pH meter [12]. For geochemical analysis, 5 tailings samples were sieved to <0.074 mm and subsequently analyzed by ICP-MS, XRF and AES. Table 2 lists the analytical method adopted for each element [13]. Certified reference materials and tailings samples were simultaneously analyzed for quality control. Four geochemical reference rock samples (GSR-2, GSR-3, GSR-5 and GSR-14) were analyzed for every tailing sample and four gold standard samples (GAu-2a, Gau-9a, Gau-10a and Gau-11a) for every tailing sample.

**Table 2.** Analytical methods used for the determination of major and trace elements.

| Element | Detection Limit | Analytical Method | RSD % | Element | Detection Limit | Analytical Method | RSD % |
|---|---|---|---|---|---|---|---|
| Ag | 0.02 mg/kg | AES | 3.82 | $K_2O$ | 0.01% | XRF | 2.2 |
| $Al_2O_3$ | 0.05% | XRF | 1.11 | MgO | 0.05% | XRF | 1.55 |
| As | 0.5 mg/kg | HG-AFS | 2.68 | Mn | 10.0 mg/kg | XRF | 0.49 |
| Au | 0.2 µg/kg | AR-GFAAS | 6.40 | Mo | 0.24 mg/kg | DF-ICP-MS | 4.76 |
| B | 1.0 mg/kg | AES | 8.18 | $Na_2O$ | 0.02% | XRF | 5.65 |
| Ba | 3.0 mg/kg | XRF | 4.30 | Ni | 2.0 mg/kg | DF-ICP-MS | 4.54 |
| Bi | 0.04 mg/kg | DF-ICP-MS | 8.85 | P | 10.0 mg/kg | XRF | 1.21 |
| C | 0.04% | VOL | 1.60 | Pb | 1.0 mg/kg | XRF | 0.62 |
| CaO | 0.05% | XRF | 0.91 | S | 50.0 mg/kg | XRF | 1.21 |
| Cd | 0.03 mg/kg | DF-ICP-MS | 5.58 | Sb | 0.04 mg/kg | DF-ICP-MS | 0.62 |
| Co | 0.5 mg/kg | DF-ICP-MS | 4.41 | $SiO_2$ | 0.05% | XRF | 4.03 |
| Cr | 5.0 mg/kg | XRF | 1.77 | Ti | 10.0 mg/kg | XRF | 0.64 |
| Cu | 0.5 mg/kg | DF-ICP-MS | 4.85 | Tl | 0.006 mg/kg | DF-ICP-MS | 2.23 |
| F | 96.0 mg/kg | ISE | 6.12 | V | 4.0 mg/kg | XRF | 9.97 |
| $Fe_2O_3$ | 0.01% | XRF | 0.65 | W | 0.3 mg/kg | DF-ICP-MS | 6.50 |
| Hg | 0.5 µg/kg | HG-AFS | 2.18 | Zn | 3.0 mg/kg | XRF | 2.23 |

Note: RSD, relative standard deviation; decomposition methods: AR, digestion with aqua regia; DF, digestion with mixed acids of $HCl + HNO_3 + HClO_4 + HF$; HG, hydride generation.

Sequential extraction was used to understand the conditions under which the PHEs in the tailings can be mobilized [14]. Every tailing sample from every drill core was analyzed by using the five-step sequential extraction method for water-soluble adsorbed exchangeable carbonate (AEC), Fe (oxy)hydroxides, Fe oxides and sulfides. The water-soluble fraction attacks sulfate minerals, such as gypsum. The considerable neutralization potential of calcite, which is extracted in the second step, makes its detection important. Besides calcite, the AEC fraction extracts the adsorbed and exchangeable ions and secondary vermiculite. The third extraction step (Fe (oxy)hydroxides fraction) attacks the secondary amorphous Fe (oxy)hydroxides and $MnO_2$. In addition to the primary oxide minerals, the fourth extraction step (Fe oxides fraction) attacks the residual secondary Fe (oxy)hydroxides of the previous extraction step. The fifth and final extraction step (sulfide fraction) attacks the primary sulfide minerals along with organics and secondary Cu sulfides. The sequential extraction method used is described by Parviainen and Loukola-Ruskeeniemi and, in more detail, by Parviainen [12]. The decanted supernatants and blank samples after each extraction step were analyzed for Cu, Pb, Zn, W, Mo, Cd, Tl, Ag, As, Bi and Hg using ICP-MS (Thermo Xseries 2). Five reagent blank samples and all samples in duplicate were analyzed to evaluate the quality and repeatability of the analytical results.

### 4. Results and Discussions

#### 4.1. Analysis of Mineral Composition Characteristics of Tailings

The mineral composition of tailings is the basic data for studying the physical and chemical properties and element occurrence state of tailings, recovering useful components of tailings, developing building materials and environmental risk assessment [15]. In order to find out the mineral composition of tailings, representative tailings samples were selected

in each mining area, and the full spectrum fitting phase semi-quantitative analysis was carried out to obtain the main mineral content data of tailings in each mining area (Table 3). The results showed that due to the differences in geological and metallogenic background and ore types, the mineral composition and content of tailings in each study area were very different. The quartz was the most abundant mineral in the tailings, with a content range of 18%–86%. The content of augite in copper mine tailings was equivalent to that of quartz, reaching up to 21%. Similarly, the quartz and augite were the top two minerals in the lead–zinc tailings, reaching up to 32% and 30%, respectively. In addition, except for gold mine tailings, there were metal minerals to be found in the other tailings. For example, 1% pyrite was found in molybdenum mine tailings, and moreover, there were 26%, 3% and 17% andradite in iron mine, lead–zinc and copper mine tailings, respectively. These metal minerals may further be acidified, leading to a risk increase of releasing heavy metals. Minerals containing more than 20% in tailings could be recycled, such as calcium andradite and augite in iron ore tailings, as well as quartz and augite in copper mine tailings. Calcite had more active chemical properties and was a good acidification neutralizing substance. It could prevent the activation and release of heavy metal elements due to acid wastewater in the tailings pond of metal sulfide deposit, which was an important index for environmental risk assessment of tailings pond. Calcite was found in iron ore tailings, lead-zinc tailings and copper mine tailings, with the contents of 4%, 14% and 11%, respectively. These results were different from that reported by Pan [2], Gao [16], Rösner [17], Zhao [18] and Chen [19]. Due to the fine particle size and large surface activity of tailings, those with high carbonate mineral content could be used to neutralize the acid water produced in the process of mining, beneficiation and metallurgy so as to realize the comprehensive treatment of the mine environment according to local conditions.

**Table 3.** Semi-quantitative analysis results of different tailings in typical mining areas of Tibet Autonomous Region.

| Type | Quartz | Andradite | Actinolite | Augite | Gypsum | Muscovite | Clinochlore | Microcline | Albite | Calcite | Pyrite | Kaolinite |
|------|--------|-----------|------------|--------|--------|-----------|-------------|------------|--------|---------|--------|-----------|
| Iron ore tailings (1#) | 18% | 26% | 19% | 21% | 1% | 1% | 2% | 2% | 5% | 4% | - | - |
| Copper mine tailings (2#) | 22% | 17% | 5% | 22% | - | 4% | 6% | 4% | 8% | 11% | - | - |
| Molybdenum tailings (4#) | 86% | - | 1% | - | - | 2% | - | 5% | 3% | - | 1% | 1% |
| Gold mine tailings (5#) | 67% | - | 3% | - | - | 12% | - | 7% | 10% | - | - | 2% |
| Lead and zinc tailings (7#) | 32% | 3% | 13% | 30% | - | <1% | 6% | 1% | 2% | 14% | - | - |

The mineral composition characteristics of different tailings samples from typical mining areas in the Tibet Autonomous Region were further studied by scanning electron microscopy. Due to the difference in geological and metallogenic background, ore type and content, there were great differences in the electron microscope photos of different tailings samples; Figure 2. For example, molybdenum mine tailings showed irregular granular shape, gold mine tailings showed irregular massive shape, and iron ore showed a cotton floc shape bonded together. The results were similar to reports by Pan [2], Zhao [18], Chen [19], Zeng [20] and Ince [21].

### 4.2. Characteristic Analysis of Element Content in Tailings

The element composition of tailings inherits the characteristics of selected ores to a large extent. At the same time, after dressing and stacking for a long time, the element content of tailings would change. In the following, we discuss the element content characteristics of tailings according to mineral classification. Due to the diverse types of non-ferrous metals in China, the complex metallogenic geological background and genetic types of ore deposited, as well as the differences in mineral processing technology and level, the element content of tailings produced by various non-ferrous metals mines varies greatly. No matter the type of tailings, the main element composition of tailings was only Si, Al, Fe, Mg, Mn, Ca, Na, K, etc., but their contents varied greatly in different tailings

(Table 4). The content of SiO$_2$ ranged from 33.07% to 91.41%, with the biggest difference. The contents of Al$_2$O$_3$, Fe$_2$O$_3$ and CaO in different types of tailings were 2.72%–15.88%, 1.94%–29.74% and 0.54%–24.70%, respectively. It was worth noting that S existed in all five types of tailings in typical mining areas, with a content of 0.17%–2.23%. The S in the tailings mainly exists as metal sulfide and easily forms acidic wastewater in the oxidizing environment, which enhances the activation dissolution rate of heavy metal elements and causes environmental pollution. The content of the main elements of tailings had a guiding significance for the development of tailings building materials. For tailings with SiO$_2$, Al$_2$O$_3$ and CaO as components, different building materials can be obtained when the ratio of tailings is different. These results were different from reports by Pan [2], Gao [16], Rösner [17], Zhao [18] and Chen [19].

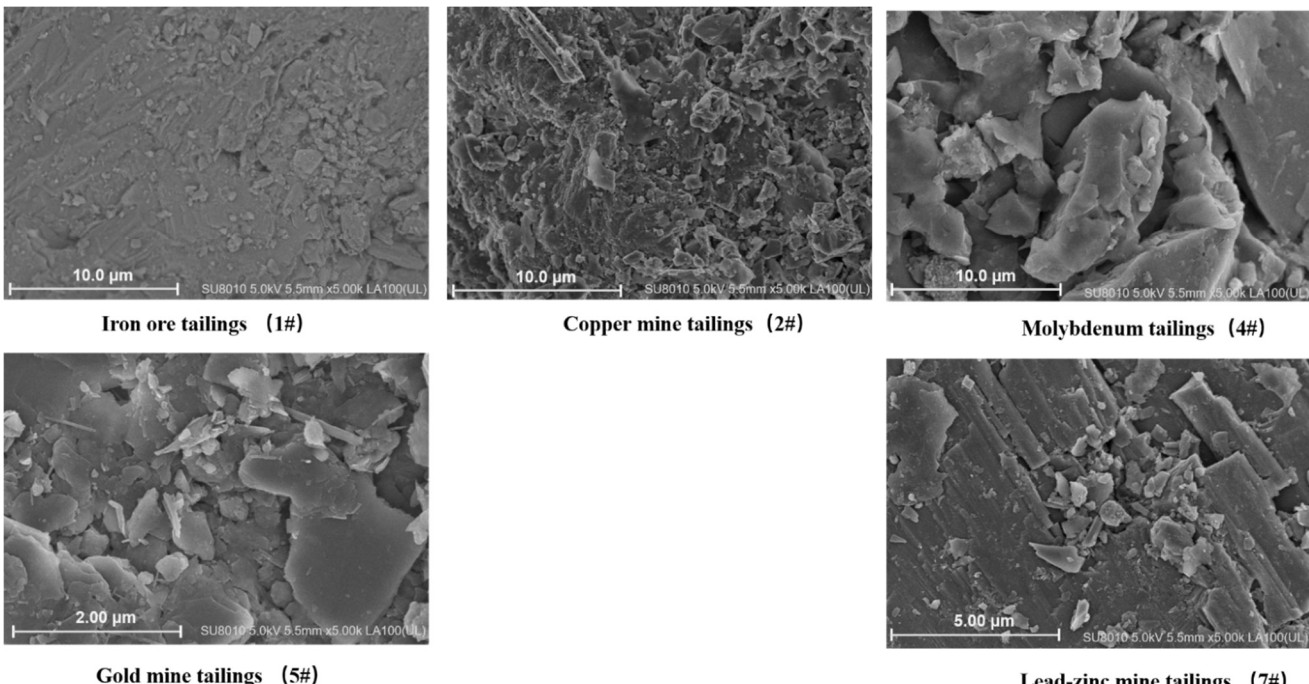

**Figure 2.** Photomicrographs of different tailings in typical mining areas of Tibet Autonomous Region.

**Table 4.** Characteristics of main elements of different tailings in typical mining areas of Tibet Autonomous Region.

| Tailing Type | The Main Elements (wt%) | | | | | | | | | | |
|---|---|---|---|---|---|---|---|---|---|---|---|
| | SiO$_2$ | Al$_2$O$_3$ | Fe$_2$O$_3$ | TiO$_2$ | MgO | CaO | Na$_2$O | K$_2$O | P$_2$O$_5$ | MnO | S |
| Iron ore tailings (1#) | 33.07 | 5.07 | 29.74 | 0.29 | 2.83 | 24.70 | 0.44 | 0.57 | 0.11 | 0.79 | 0.94 |
| Copper mine tailings (2#) | 41.97 | 9.28 | 19.13 | 0.52 | 2.05 | 23.45 | 0.74 | 1.34 | 0.16 | 0.41 | 0.17 |
| Molybdenum tailings (4#) | 91.41 | 2.72 | 1.94 | 0.13 | 0.44 | 0.54 | 0.44 | 1.62 | 0.09 | 0.04 | 0.47 |
| Gold mine tailings (5#) | 67.61 | 15.88 | 5.87 | 0.48 | 1.06 | 1.43 | 0.89 | 5.78 | 0.27 | 0.09 | 0.17 |
| Lead–zinc mine tailings (7#) | 41.80 | 4.44 | 23.73 | 0.21 | 1.72 | 20.65 | 0.23 | 0.29 | 0.13 | 3.20 | 2.23 |

### 4.3. Characteristic Analysis of Trace Elements in Tailings

The content of trace elements in tailings was closely related to mineralization, which was characterized by a high content of ore-forming elements and associated elements, some of which had reached the lowest industrial grade and potential recycling value [2]. At the same time, the high content of heavy metal elements made the tailings pond a potential environmental pollution source [2]. The characteristics of trace elements in different tailings of typical mining areas of the Tibet Autonomous Region are shown in Table 5. The contents of Mn, Zn and Ni in iron ore tailings from typical mining areas in the

Tibet Autonomous Region were 0.611%, 0.407% and 0.438%, respectively, which exceed the minimum industrial grade and, thus, have potential utilization value. The contents of Mn and Zn in the lead–zinc tailings were 2.48% and 0.825%, respectively, and the contents of Pb and P were also high, both of which were 0.156%, indicating that the potential utilization value was high. The contents of Mn and Cu in copper tailings were 0.317% and 0.246%, respectively, and the contents of Ni also reached 0.162%. These higher elements had better utilization value. It was worth noting that Mo was not detected in any other type of tailings except molybdenum tailings with a content of 0.018%. In addition, it could be seen from Table 4 that the distribution of trace elements was extremely uneven, and local enrichment may become an available orebody. These results were different from those reported by Pan [2], Gao [16], Zhao [18], Kan [22] and Fontes [23].

**Table 5.** Characteristics of trace elements in different tailings of typical mining areas in Tibet Autonomous Region.

| Tailing Type | Trace Elements (wt%) | | | | | | | | | | |
|---|---|---|---|---|---|---|---|---|---|---|---|
| | Mn | Zn | As | Pb | Cu | Ni | P | Co | Mo | Y | Rb |
| Iron ore tailings (1#) | 0.611 | 0.407 | 0.010 | - | 0.078 | 0.438 | 0.047 | 0.014 | - | 0.005 | 0.006 |
| Copper mine tailings (2#) | 0.317 | 0.028 | 0.005 | 0.015 | 0.246 | 0.162 | 0.068 | 0.011 | - | 0.003 | 0.005 |
| Molybdenum tailings (4#) | 0.029 | 0.018 | - | 0.050 | 0.001 | 0.001 | 0.04 | - | 0.018 | - | 0.003 |
| Gold mine tailings (5#) | 0.065 | 0.022 | - | 0.229 | 0.022 | - | 0.118 | 0.003 | - | 0.003 | 0.003 |
| Lead–zinc mine tailings (7#) | 2.48 | 0.825 | 0.016 | 0.156 | 0.011 | 0.006 | 0.156 | 0.012 | - | 0.002 | 0.002 |

*4.4. Characteristics Analysis of Radioactive Elements in Tailings*

Most of the tailings of different types contain more than two kinds of minerals or a variety of common and associated useful elements or components. In the process of exploitation and utilization of mineral resources, radioactive elements would be concentrated in the tailing sand, which could increase the natural radioactivity level and radioactive pollution in the surrounding environment and bring potential harm to environmental safety and human health. Table 6 showed the characteristics of radioactive elements in different tailings of typical mining areas in Tibet Autonomous Region. Radioactive elements (RA-226, TH-232 and K-40) shared by different tailings of typical mining areas in the Tibet Autonomous Region were detected. It was found that each radioactive element had a great difference in different types of tailings. The specific activity of Ra-226 was the highest in molybdenum tailings, reaching 94.62 Bq/Kg, and the lowest in iron ore tailings, reaching 30.18 Bq/Kg. The specific activity of Th-232 was the highest in the gold mine, reaching 89.62 Bq/Kg, and the lowest in molybdenum mine tailings, reaching 16.09 Bq/Kg. The specific activity of K-40 was the highest in the gold mine, which was 762.89 Bq/Kg, and the lowest in lead–zinc tailings, which was 19.44 Bq/Kg. These results were different from reports by Pan [2], Kan [22], Fontes [23], Ngole-Jeme [24] and Wu [25].

**Table 6.** Characteristics of radioactive elements in different tailings of typical mining areas in Tibet Autonomous Region.

| Tailing Type | Radioactive Element (Bq/Kg) | | | | |
|---|---|---|---|---|---|
| | Ra-226 Specific Activity | Th-232 Specific Activity | K-40 Specific Activity | Internal Exposure Index (IRa) | External Exposure Index (Ir) |
| Iron ore tailings (1#) | 30.18 | 21.58 | 86.95 | 0.2 | 0.2 |
| Copper mine tailings (2#) | 34.42 | 27.26 | 89.65 | 0.2 | 0.2 |
| Molybdenum tailings (4#) | 94.62 | 16.09 | 369.66 | 0.5 | 0.4 |
| Gold mine tailings (5#) | 83.88 | 89.62 | 762.89 | 0.4 | 0.8 |
| Lead–zinc mine tailings (7#) | 31.70 | 18.81 | 19.44 | 0.2 | 0.2 |

*4.5. Analysis of Particle Size Characteristics of Tailings*

Tailings samples were tested using a laser particle size distribution apparatus. The suspension was water, and the sample was dispersed for 3 min by ultrasonic wave. Then, the sample was continuously measured three times and a stable value was obtained. Figure 3 shows that the particle size distribution of tailings of different typical mining areas in the Tibet Autonomous Region presented different degrees of difference. When the size of molybdenum tailings was 211 μm, the volume of molybdenum tailings reached the maximum, which was 5.96%. When the particle size of gold tailings was 1430 μm, the volume of gold tailings reached the maximum, which was 8.94%. When the particle size of iron ore tailings was 586 μm, the volume reached the maximum, which was 5.92%. The volume of lead–zinc tailings reached a maximum of 5.41% when the particle size was 3,101,430 μm, while the volume of copper tailings reached a maximum of 8.15% when the particle size was 86.4 μm. These results were different from those of Pan [2], Gao [16], Ngole-Jeme [23], Fontes [24] and Wu [25].

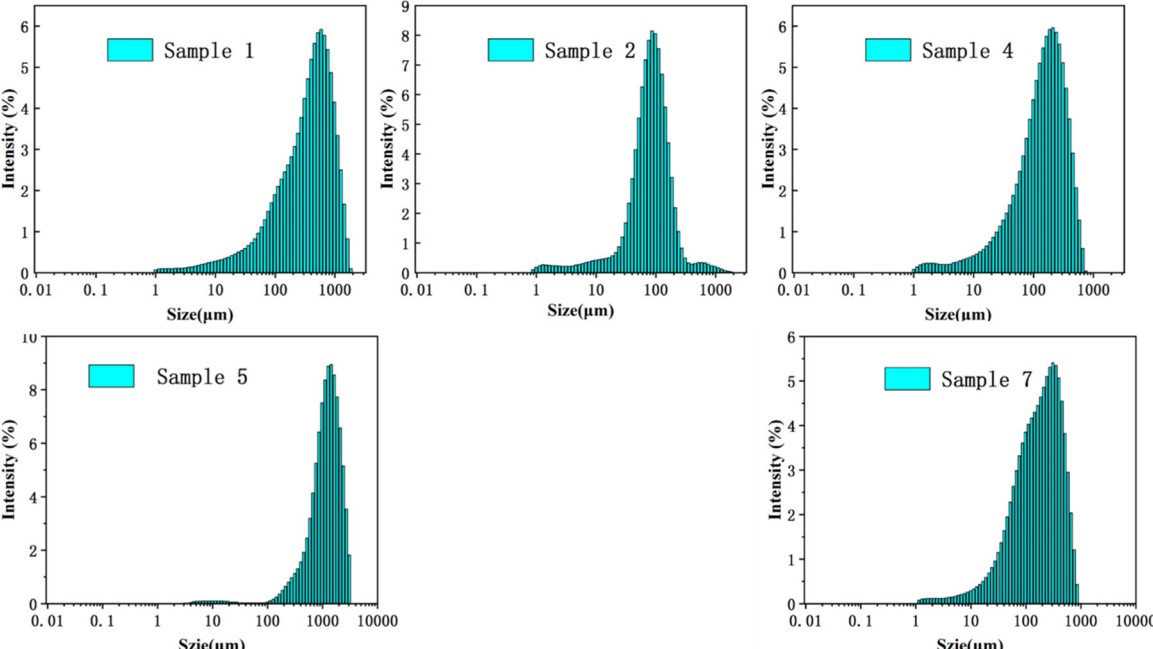

**Figure 3.** Grain size distribution of different tailings in typical mining areas of Tibet Autonomous Region. Iron ore tailings (1); Copper mine tailings (2); Molybdenum tailings (4); Gold mine tailings (5); Lead–zinc mine tailings (7).

## 5. Conclusions

The mineral composition, chemical composition, elemental composition and grain size of five tailings samples from typical metal mining areas in the Tibet Autonomous Region were analyzed by means of elemental earth survey, mineralogy and environmental geochemistry. The results showed that the mineral composition of different tailings samples from typical metal mining areas in the Tibet Autonomous Region was complex and diverse; the content was also greatly different. Quartz was the main mineral found in all tailings. The main elements in the tailings reservoir samples were $SiO_2$, CaO, Fe, $Al_2O_3$ and S. Although after beneficiation, the content of ore-forming and associated elements in tailings was still generally high, such as Mn, Zn, Cu, Ni, Pb, P, etc. Some elements were of minimal industrial taste and had potential recycling value. Each radioactive element (RA-226, TH-232 and K-40) varied greatly in the samples from different mining areas, which might bring potential harm to human health and surrounding environment safety. By analyzing the particle size of different tailings samples, it was found that the particle size of different tailings samples showed different degrees of difference. The above research results could provide

detailed and reliable technical data and a theoretical basis for the subsequent comprehensive development and utilization of tailings ponds and the protection of the surrounding ecological environment, which has important theoretical and practical significance.

**Author Contributions:** R.W.: supervision, writing—original draft, project administration, funding acquisition. X.H.: methodology, data curation, visualization. F.T. and G.C.: investigation, data curation. L.N. and L.P.: visualization, supervision. D.L.: writing—reviewing and editing, supervision. B.X.: supervision, writing—reviewing. All authors have read and agreed to the published version of the manuscript.

**Funding:** This research is financially supported by the National Key Research and Development Program of China in 2019 (No. 2019YFC1904101), the General project of Fujian Provincial Natural Science Foundation (2021J011054), the Initial Scientific Research Foundation of Fujian University of Technology (GY-Z20014), the Research Development Foundation of Fujian University of Technology (GY-Z18186).

**Data Availability Statement:** The data presented in this study are available on request from the corresponding author.

**Conflicts of Interest:** The authors declare no conflict of interest.

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
