# Peer review of "Geochemical Characteristics of Tailings from Typical Metal Mining Areas in Tibet Autonomous Region"

_minerals, doi:10.3390/min12060697_

Round 1

Reviewer 1 Report

Weng and colleagues present new tailings data from five mine locations from the Tibet Autonomous Region. Although I think this is a valuable data set it is not clear to me how they have used this data to contribute to ongoing developments in the areas of either acid mine drainage (AMD) or reprocessing of tailings material. While AMD is not a new concept, there is a growing number of recent tailings reprocessing studies that could be addressed here. Discussing the results as well as presenting them would greatly strengthen this paper. While the data appear to be of high quality and have the potential to contribute to ongoing research in tailings from the Tibet Autonomous Region, which would be of interest to the readers of Minerals, I have a number of suggestions for improvement that I have outlined below.

General comments:

This manuscript would benefit from editing and proofing by a first speaking English author to improve the literacy and grammar. There are many instances where minor improvements would greatly enhance the readability sentences and following paragraphs (e.g., lines 11, 30-31, 34, 60-61, 150-160, etc…).

What are the greater implications of the findings in this paper? It is mentioned multiple times that quartz is abundant in all tailings materials but what does this mean in terms of reprocessing, usability, potential for AMD? Is there demand for ore sands in this region? There are many papers and reports in this area, e.g., “Golev et al., 2002: Ore-sand: A potential new solution to the mine tailings and global sand sustainability crises”. Concluding this paper with connections to current literature and case studies will make for a stronger paper. As it stands it is not clear how this paper is advancing research in these areas.

The same rational and review should be considered to those potential recoverable elements mentioned in this paper. Is there demand for a certain commodity in this region? What are the minable grades of some of these elements that are considered high in concentration from these tailings? Was a greater range of elements measured but not reported? Again, making connections to current literature and case studies will make for a stronger discussion and conclusion.

In addition to making connections with the potential recoverable materials in the tailings comparing the size and volume of the tailing’s dams studied to other sites is an important factor to consider. Adding these parameters to site descriptions would help the reader understand the potential for both recovery and potential for AMD.

Specific comments as follow:

Line 15 – Do you mean important or abundant?

Line 73- 74 – Addition of sampling depths would be good to understand distribution of samples collected.

Line 82 – Correct the spacing in figure caption.

Line 96 – This paragraph would benefit with more details, what rig was used to sample, what were the maximum sampling depths reached at each site, any initial visual observations that were made, oxidation at surface, colour changes, etc…;

Line 100 – Do you mean? Reword this, each sample weighed between 2.5kg to xx?

Line 104 – Were the samples chosen a 1m sample intervals alone or were these composites from each site? What was the rationale behind choosing only one sample per site? Why were different intervals used for different testing?

Line 113 – Why were the samples sieved? Excluding coarser particle size could naturally exclude certain minerals.

Line 119 – Can this caption fit on the next page with the table to improve readability?

Line 150 – Do you mean abundant instead of important?

Line 150-153 – Improve reporting of mineral concentrations to improve readability.

Line 153-154 – Rephrase and give examples of what you mean by ‘metal minerals’.

Line 154- 156 – Rephrase sentence.

Line 158-160 – Rephrase sentence.

Line 167-170 – This would depend on other mineral contributions in the tailings, do you have any examples?

Line 185-195 – If these sentences do not relate directly to results authors should consider moving this text to the introduction.

Line 207 – How were the results different to the other papers cited?

Line 210 – Introduce Table 4 at the first use of the values from it in this section. For example, at line 216.

Line 222 – Which elements have better utilisation value?

Line 226-227 – Again how do these results differ to these other papers cited?

Line 249-251 – The first two sentences here should go into methods.

Line 261 – Again, how to these results differ to the papers cited?

Line 275 – Which elements have low recycling value? It helps to be specific in the conclusion.

Line 279-282 – See general comments. There is data within this paper that can comment on the general theme mentioned in this final sentence. Using the data collected here against other example will strength this conclusion and the importance of this paper as a stand alone piece.

Tables and Figures:

Figure 3 - These graphics need improvement; they appear fuzzy and the numbers on both the X and Y axis are hard to read. Increasing font size would help the readability of this figure.

Table 3 – Are there no other sulfides present? Chalcopyrite, galena, sphalerite?

Author Response

Dear Editors and Reviewers:

        Thank you for your letter and for the reviewers’ comments concerning our manuscript entitled“Geochemical characteristics of tailings from typical metal mining areas in Tibet Autonomous Region”(minerals-1738351). Those comments are all valuable and very helpful for revising and improving our paper, as well as the important guiding significance to our research. We have studied the comments carefully and have made corrections which we hope meet with approval. Revised portions are marked in red on the paper. The main corrections in the paper and the responses to the reviewer’s comments are as flowing:

 Responds to the review’s comments:

1. This manuscript would benefit from editing and proofing by a first speaking English author to improve the literacy and grammar. There are many instances where minor improvements would greatly enhance the readability sentences and following paragraphs (e.g., lines 11, 30-31, 34, 60-61, 150-160, etc…).

Response: We are very sorry for our incorrect writing. The inappropriate sentences and grammar you said have made careful correction (e.g., lines 11, 30-31, 34, 60-61, 150-160, etc   ). Revised portion are marked in red in the paper.

2. What are the greater implications of the findings in this paper? It is mentioned multiple times that quartz is abundant in all tailings materials but what does this mean in terms of reprocessing, usability, potential for AMD? Is there demand for ore sands in this region? There are many papers and reports in this area, e.g., “Golev et al., 2002: Ore-sand: A potential new solution to the mine tailings and global sand sustainability crises”. Concluding this paper with connections to current literature and case studies will make for a stronger paper. As it stands it is not clear how this paper is advancing research in these areas.

Response: Thank you for your valuable comments. We reported the overall and systematic geochemical investigation on the elements of tailings ponds of different metals in the Tibet Autonomous Region, including the mineralogy, chemical composition, element composition and particle size characteristics of tailings. These results are of great significance to the future utilization and resource utilization of tailings pond in Tibet Autonomous Region. Abundant quartz minerals showed the mineral composition characteristics of tailings in Tibet Autonomous Region. The results are helpful to the recovery of useful components of tailings, the development of building materials and the environmental risk assessment. In our study, pyrite in metal sulfide is the main mineral that produces acid wastewater. The result is helpful analysis the formation of acid water (AMD). We have also paid attention to the potential new solutions to the sustainability crisis of mineral sands: tailings and global sands reported by Golev et al., 2002. The analysis of mineral composition and element analysis in our study can provide some basic data for solving these crises.

3. The same rational and review should be considered to those potential recoverable elements mentioned in this paper. Is there demand for a certain commodity in this region? What are the minable grades of some of these elements that are considered high in concentration from these tailings? Was a greater range of elements measured but not reported? Again, making connections to current literature and case studies will make for a stronger discussion and conclusion.

Response: Considering the reviewer’ suggestion, we have looked for exploitable grades of certain elements in these tailings that were considered to be high concentrations. The minimum industrial grades of Mn, Zn and Cu are 0.3%, 0.1% and 0.4%. With the implementation of China's western development, there is demand for a certain commodity in the Tibet Autonomous Region. A greater range of elements had measured, but the content of these elements is very low or below the detection limit (e.g., Ce, Cr, Sn, etc   ). In our research, we had compared our experimental results with others (e.g., Pan, Gao, Zhao, Kan, Fontes, etc   ) and got different research results.

4. In addition to making connections with the potential recoverable materials in the tailings comparing the size and volume of the tailing’s dams studied to other sites is an important factor to consider. Adding these parameters to site descriptions would help the reader understand the potential for both recovery and potential for AMD.

Response: We are very sorry for our negligence of the tailings comparing the size and volume of the tailing’s dams studied to other sites. These experimental data were not found in this study, and we will focus on this issue in the next study.

Specific comments as follow:

Line 15 – Do you mean important or abundant?

Response: Abundant

Line 73- 74 – Addition of sampling depths would be good to understand distribution of samples collected.

Response: We have added the sampling depth in our manuscript.

Line 82 – Correct the spacing in figure caption.

Response: We have corrected the spacing in figure caption.

Line 96 – This paragraph would benefit with more details, what rig was used to sample, what were the maximum sampling depths reached at each site, any initial visual observations that were made, oxidation at surface, colour changes, etc…;

Response: Drilling machines (diesel rigs) were used to collect tailings cores. The maximum sampling depth was 10 m. The color of all samples is mainly taupe.

Line 100 – Do you mean? Reword this, each sample weighed between 2.5kg to xx?

Response: Good suggestion, each sample weighed between 2.5 kg to 3.0 kg

Line 104 – Were the samples chosen a 1m sample intervals alone or were these composites from each site? What was the rationale behind choosing only one sample per site? Why were different intervals used for different testing?

Response: The samples were chosen 1m sample intervals alone. This sampling method is adopted in most literatures.

Line 113 – Why were the samples sieved? Excluding coarser particle size could naturally exclude certain minerals.

Response: Samples were screened to study typical metal tailings in the Tibet Autonomous Region

Line 119 – Can this caption fit on the next page with the table to improve readability?

Response: Very nice suggestion. We have corrected the portion.

Line 150 – Do you mean abundant instead of important?

Response: Yes, we have corrected the portion.

Line 150-153 – Improve reporting of mineral concentrations to improve readability.

Response: We have corrected the portion. Thank you!

Line 153-154 – Rephrase and give examples of what you mean by ‘metal minerals’.

Response: We have corrected the portion. Thank you!

Line 154- 156 – Rephrase sentence.

Response: We have rephrased sentence.

Line 158-160 – Rephrase sentence.

Response: We have rephrased sentence.

Line 167-170 – This would depend on other mineral contributions in the tailings, do you have any examples?

Response: The Similar conclusions had been described by Pan et al., 2002 in the reference 2.

Line 185-195 – If these sentences do not relate directly to results authors should consider moving this text to the introduction.

Response: Some sentences have moved to the introduction, others have survived for explaining the experimental results. The detailed description is displayed in the manuscript.

Line 207 – How were the results different to the other papers cited?

Response: For example, the content of SiO2, Al2O3 and Cao was 64.2%, 44.6%, 55.7%, respectively. The content of Al2O3 was 16.2%, 32.6%, 25.9%, respectively. The content of Cao was 2.3%, 17.4%, 31.8%, respectively, reported by Pan [2], Gao [16], Rösner [17], Zhao [18], and Chen [19]. These results were different reported in our study.

Line 210 – Introduce Table 4 at the first use of the values from it in this section. For example, at line 216.

Response: We have added the sentence in this section:“The Characteristics of trace elements in different tailings of typical mining areas of Tibet Autonomous Region was shown on the Table 5.”

Line 222 – Which elements have better utilisation value?

Response: Mn, Zn, Ni, Cu, Pb and P have better utilisation value. Because the content of these elements was higher than that of other elements.

Line 226-227 – Again how do these results differ to these other papers cited?

Response: For example, the content of Cu was 0.00045%, 0.0051%, 0.00078%, respectively, reported by Pan [2], Kan [22], and Fontes [24]. These results were different reported in our study.

Line 249-251 – The first two sentences here should go into methods.

Response: Very good advice. We have moved the first two sentences to methods.

Line 275 – Which elements have low recycling value? It helps to be specific in the conclusion.

Response: As, Co, Mo and Rb have low recycling value. Because the content of these elements was lower than that of other elements.

Line 279-282 – See general comments. There is data within this paper that can comment on the general theme mentioned in this final sentence. Using the data collected here against other example will strength this conclusion and the importance of this paper as a stand alone piece.

Response: Thank you for your valuable comments. The above research results could provide detailed and reliable technical data and theoretical basis for the subsequent comprehensive development and utilization of tailings pond and the protection of surrounding ecological environment, which has important theoretical and practical significance.

Tables and Figures:

 图3 - 这些图形需要改进;它们看起来很模糊,X轴和Y轴上的数字都很难阅读。增加字体大小将有助于此图的可读性。

响应:我们重新更改了所有图片中的字体大小,并希望满足发布要求。

表3 – 是否存在其他硫化物?黄铜矿、方铅矿、闪锌矿?

响应:结果表明,除了表3上的黄铁矿外,没有黄铜矿,方铅矿和闪锌矿。可能的原因是这些矿石的含量低于检测限值。

Reviewer 2 Report

Dear authors,

the paper is well written and present many types of analyses.

I recommend to present in Fig 1 and to describe in Table 1, those 7 sampling points that you mentioned, otherwise you can rename and mention only the points from which you have data. Table 1 presents "dimension", but with no unit to have an idea about what you refer to.

Review line 97;

For Figure 2 the scale is not visible; Fig 3 is not clear and the present quality is not proper for publishing; as well the figure do not present the samples in a order.

For Materials and Method section, the detailed samples preparation for each analyses method  (and references) must be added.

In Table 5, the trace elements are presented as percent?

Lines 272 - use the subscript function.

Review all the manuscript and clarify the term of " heavy metals".

Good luck with the paper!

Author Response

Dear Editors and Reviewers:

Thank you for your letter and for the reviewers’ comments concerning our manuscript entitled“Geochemical characteristics of tailings from typical metal mining areas in Tibet Autonomous Region”(minerals-1738351). Those comments are all valuable and very helpful for revising and improving our paper, as well as the important guiding significance to our researches. We have studied comments carefully and have made correction which we hope meet with approval. Revised portion are marked in red in the paper. The main corrections in the paper and the responds to the reviewer’ comments are as flowing:

      Responds to the review’ comments:

1. I recommend to present in Fig 1 and to describe in Table 1, those 7 sampling points that you mentioned, otherwise you can rename and mention only the points from which you have data. Table 1 presents "dimension", but with no unit to have an idea about what you refer to.

Response: Thank you for your valuable comments. The corresponding positions (in Table 1) of the five sampling points have been marked in Fig 1. Types of 5 tailings ponds were shown in figure caption. Altitude, longitude and dimension showed the specific location of the sampling point.

2. Review line 97;

Response: In this study, 17 tailings samples were collected from 17 drill cores. We selected five typical metal mining areas from 17 samples to study the elemental geochemical characteristics of tailings from typical metal mining areas in Tibet Autonomous Region.

3. For Figure 2 the scale is not visible; Fig 3 is not clear and the present quality is not proper for publishing; as well the figure do not present the samples in a order.

Response: We have made correction for Fig 2 and Fig 3 according to the reviewer’ comments. Revised portion were shown in the manuscript.

4. For Materials and Method section, the detailed samples preparation for each analyses method (and references) must be added.

Response: We have made correction according to the reviewer’ comments.

5. In Table 5, the trace elements are presented as percent?

Response: Yes, the trace elements are presented as percent In some related literatures (such as Pan et al., 2002).

6. Lines 272 – use the subscript function

Response: We are sorry for our negligence of the portion (Al2O3).

7. Review all the manuscript and clarify the term of " heavy metals"

Response: Considering the reviewer’ suggestion, we have carefully review all the manuscript. The definition of" heavy metals" is metals with a density greater than 4.5 g/cm3, including gold, silver, copper, iron, mercury, lead, cadmium, etc. The accumulation of heavy metals in the human body to a certain extent will cause chronic poisoning. In this study, the heavy metals we studied include Mn, Zn, Cu, Pb and so on.
